# Parity Check Based Fault Detection against Timing Fault Injection Attacks

Maoshen Zhang [1], He Li [2], Peijing Wang [1] and Qiang Liu [1,*]

1   School of Microelectronics, Tianjin University, Tianjin 300072, China
2   School of Electronic Science and Engineering, Southeast University, Nanjing 210096, China
*   Correspondence: qiangliu@tju.edu.cn

**Abstract:** Fault injection technologies can be utilized to steal secret information inside integrated circuits (ICs), and thus cause serious information security threats. Parity check has been adopted as an efficient method against fault injection attacks. However, the contradiction between security and overhead restricts the further development and applications of parity check in fault injection detection. This paper proposes two methods, mixed-grained parity check and word recombination parity check, based on parity check for the trade-off between security and overhead. The efficiency of the proposed approaches is verified on RC5, AES, and DES encryption implementations by clock glitch attack. Compared with the traditional parity check, the fault coverage rate of the mixed-grained approach can be increased by up to 53.69% by consuming 13.2% registers more. Against the traditional parity check, the fault coverage rate of the word recombination approach can be increased by up to 47.16% by using only 2.35% register more. The proposed approaches provide IC designers with countermeasure options targeting different design skills and design specifications.

**Keywords:** fault injection; information security; parity check

## 1. Introduction

The Internet of things (IoT) has been prevailing in a wide range of applications, including the smart home, smart transportation, smart city, etc. The distributed and connected IoT devices in these applications continuously interact with personal data or sensitive information, and face great security threats [1].

In addition to widely known software attacks, hardware physical attacks, such as side-channel analysis (SCA) [2] and fault injection attack (FIA) [3], on cryptographic primitives embedded in the IoT devices have become the main threat. SCA exploits the physical information, such as power trace or delay variation, of the operating IoT devices to extract the critical data such as secret keys. In contrast to the passive SCA, FIA is the active technique, which intentionally injects data faults into the IoT devices and reveals the confidential information by analyzing the faulty outputs [4]. With the active and controllable nature, FIA demonstrates higher attack efficiency and higher potential hazards [5–7]. Among the FIA techniques, the low-cost techniques, such as clock glitch attack [8], voltage glitch attack [9] and electromagnetic harmonic attack [10], aim to disturb the clock system of the IoT devices to inject timing faults. The timing faults are mainly due to violation of setup/hold timing constraints of the target circuits and induce errors in data processing. By analyzing the erroneous outputs, the confidential information can be revealed.

Two main types of countermeasures against timing FIAs have been proposed, sensor-based countermeasure [11,12] and information-based countermeasure [13,14]. Sensor-based countermeasures deploy analog/digital sensors, which are integrated into the target circuits, to detect the disturbance of the clock system. The advantage of the sensor-based countermeasure is that the fault injection can be detected even before the fault is effective. However, this type of countermeasure requires advanced circuit design skills to deal

with the issues: (1) high power consumption of analog sensors using phase locked loop, or (2) the aging phenomenon of digital sensors using ring oscillator. Information-based countermeasures exploit various error detection codes (EDCs), such as cyclic residue code and parity code, to detect the induced errors in data processing [14]. The information-based methods are applied at the information-flow level, requiring fewer circuit details and showing the independence of the particular hardware implementation.

The basic principle of fault detection in the information-based methods is that the predicted information bits (the residue for the cyclic residue code and the parity for the parity code) and the actual information bits are compared. Then, a mismatch shows the fault injection. The fault coverage is improved as the number of redundant information bits increases. Among the two simple EDCs, the parity code-based detection method shows similar fault coverage and less hardware overhead, due to its simple operations in parity prediction [14]. It is known that the parity check and its variants (e.g., Low-Density Parity Check Code) are widely used in the communication area. Ref. [13] firstly implemented the parity check on the AES cipher circuit with the uniform grain, and [14] compared the performance of the parity code and the cyclic residue code for fault detection of the RC5 cipher circuit. As we know, the parity-code-based method fails when the number of the fault bits is even [11]. Therefore, for a higher fault coverage rate, the fine-grained parity code check is exploited but introduces higher overhead in hardware resource or power consumption [15].

For the satisfactory trade-off between fault coverage and overhead, two parity-code-based detection are proposed in this paper. Firstly, a mixed-grained parity check approach is proposed, in which the sub-word level parity check (high fault coverage and overhead) is applied to the security-critical operations of the circuit, and the word-level parity check (low fault coverage and overhead) is applied to the other operations. This way achieves a trade-off between fault coverage rate and overhead. Secondly, to further reduce the overhead, a word recombination parity check approach is proposed. Sub-words of different variables from the security-critical operations and non-critical operations are reorganized into a new word. Then the word-level parity check is performed on the new word. This way is equivalent to the sub-word level parity check applied to the security-critical operations. The contributions of this work are as follows:

- We propose two efficient detection approaches against timing FIAs based on the parity code check. The two approaches realize the idea of fine-grained parity check with low overhead in two ways and apply the parity check on the pipelined and iterative circuits. The two approaches provide designers with different design capabilities with alternative countermeasures.
- We develop the implementation flow of the proposed approaches, which can be integrated with the existing IC design flow, enabling security-driven hardware design flow.
- We design parity check blocks for basic operations involved in various encryption algorithms. In this way, the proposed methods apply to multiple widely used cryptography ICs.
- We evaluate the proposed approaches on RC5, AES, and DES encryption implementations. Compared with the word parity check, the results show that the mixed-grained approach increases the fault coverage rate by up to 53.69% while consuming 13.2% more resources; the word recombination approach increases the fault coverage rate by up to 47.16% while introducing up to 2.35% resource usage.

The rest of this paper is organized as follows. Section 2 briefly introduces the background. Section 3 describes the threat model which is considered in this paper. Section 4 explains the concepts and the implementations of the proposed approaches. Section 5 shows the prediction operations for the parity check of various ciphers. Section 6 proposes the general implementation flow of the proposed approaches. Section 7 shows the experiment settings and results. Section 8 compares the related works with the proposed methods. Section 9 concludes this paper.

## 2. Background

We first introduce the principle of parity-code-based detection (Section 2.1), and then briefly describe the basic operations in three typical encryption algorithms, AES, RC5, and DES (Section 2.2).

### 2.1. Principle of Parity Code-Based Detection

The parity bit $P(A)$ of an $n$-bit word $A$ is obtained by XORing all of its $n$ bits as below

$$P(A) = \overset{i=(n-1)\sim 0}{\oplus} A[i] = A[n-1] \oplus \ldots \oplus A[0] \tag{1}$$

To implement the parity code-based detection method, as shown in Figure 1, a parity check block is added in parallel with the circuit under attack (CUA), including parity calculation, prediction and comparison. The parity code-based detection approach detects the timing fault injection by checking the calculated parity bit $P(y)$ and the predicted parity bit $P'(y)$ of the output $y$. $P(y)$ is calculated using (1). $P'(y)$ is obtained dependently on the specific operations in CUA. Section 5 will present the ways of parity prediction for basic encryption operations. The comparison is realized by XORing $P(y)$, and $P'(y)$ and generates a check bit. When a fault is injected into the CUA, $y$ and thus $P(y)$ may be disturbed. The check bit equal to 1 indicates that the fault is detected. For example, let the correct value of 32-bit output $y$ in Figure 1 be $0xFFFF$. When a timing fault in the CUA alters $y$ to $0xFFFE$, $P(y)$ turns from 0 to 1, and the predicted parity $P'(y)$ based on $x$ is still 0. Then, the check bit turns from 0 to 1, and the fault is detected.

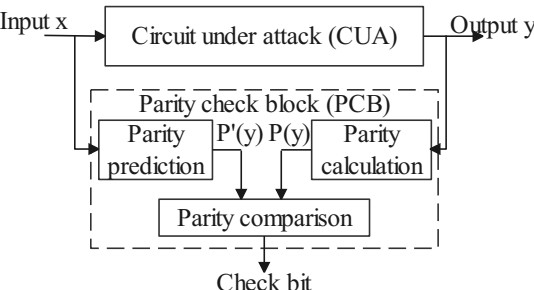

**Figure 1.** The diagram of the parity-check-based detection approach.

Parity check with $m$-bit granularity means that one parity bit is computed for every $m$ bits of a $n$-bit data, i.e., $k = \lceil n/m \rceil$ parity bits are needed. The finer granularity means the smaller $m$ and the larger $k$. Continue with the above example. Assume that a 2-bit fault is injected into the CUA, i.e., the fault induces timing violations in two paths. The fault changes $y$ to $0xFEFE$. In this case, with the 32-bit parity check, $P(y)$ remains 0 and the check bit equals 0. The fault cannot be detected. If a 16-bit parity check is applied, two parity bits are calculated and predicted. $P(y[31\ldots 16])$ and $P(y[15\ldots 0])$ become 1, respectively. Two check bits turn from 0 to 1, and the fault is detected.

This example shows that the fine-grained parity check improves fault coverage, but requires extra parity bits and hardware resources. To trade off the fault coverage and hardware overhead, this paper proposes two efficient approaches which will be presented in the next section.

### 2.2. Basic Operations in RC5, DES and AES Encryption Algorithms

Algorithms 1–3 show the basic operations involved in three widely used symmetric block ciphers, RC5, DES, and AES. Among them, RC5 and DES use data-dependent operations, while AES uses data-independent operations. DES is a typical Feistel structure cipher. AES is a typical substitution-permutation network (SPN) structure cipher. All three ciphers involve $r$-round (iterative) operations. The basic operations of the three encryption algorithms are also common for some other encryption algorithms such as IDEA [16] and

Twofish [17]. This paper presents and evaluates the proposed approaches with these three encryption algorithms.

---

**Algorithm 1** RC5 algorithm

---

**Input** 64-bit plaintext, which is divided into 32-bit words A, B, and key.
**Output**: ciphertext
1: A, B = Permutation-P(plaintext)
2: A=A+key [0]
3: B=B+key [1]
4: for i =1 to 12
5:      A = ((A xor B) $<<<$B)+key[2i]
6:      B = ((B xor A)$<<<$A)+key[2i+1]
7: ciphertext = {A, B}.

---

**Algorithm 2** DES algorithm

---

**Input** 64-bit plaintext divided into 32-bit *L* and *R*, and key.
**Output:** ciphertext
**Begin**
1:   for i = 1 to 16
2:       $L_i = R_{i-1}$;
3:       $R_i = L_{i-1}$ xor function-f $(R_{i-1}, key_i)$;
4:   ciphertext = {$L_{16}$, $R_{16}$};**End**
**Function-f(R, key)**
5:   E = Expansion-E ( R )
6:   Address_Sbox = E xor key
7:   S = Sbox (Address_Sbox)
8:   P = Permutation-P (S).

---

**Algorithm 3** AES-128 algorithm

---

**Input:** 128-bit Plaintext, which is presented as the state matrix, and key.
**Output:** ciphertext
**Begin:**
1:    state = Plaintext
2:    AddRoundKey (state, key)
3:    for round=1 to 9
4:        state=Subbytes(state)
5:        state=shiftRows(state)
6:        state=MixColumns(state)
7:        state=AddRoundKey(state, key[round]). **End**
**Final round Begin:**
8:    state=Subbyte(state)
9:    state=shiftRows(state)
10:    state=AddRoundKey(state, key[round])
11:    ciphertext=state. **End**

---

As shown in Algorithm 1, each round of RC5 is comprised of three basic operations: XOR, rotate left ($<<<$), and arithmetic addition (+). In Algorithm 2, each round of DES contains four basic operations: Expansion permutation box (EP-box), XOR, Substitution boxes (S-box) and Straight permutation box (SP-box). The EP-box operation expands the 32-bit input to 48-bit by a permutation table with $8 \times 6$ entries, each corresponding to one bit of the input. The S-box operation carries out the data mixing using eight S-box tables, each with $4 \times 16$ 4-bit entries. The SP-box operation performs straight permutation with a $4 \times 8$ table, each corresponding to one bit of the input. In Algorithm 3, each round of AES contains four basic operations: SubBytes, ShiftRows, MixColumns, and AddRoundKey. The SubBytes operation substitutes the state matrix (16 bytes) by looking up a fixed table

(S-box). The ShiftRows operation rotates each of the four rows of the matrix to the left. The *j*-th row is shifted $j-1$ (byte) position. The MixColumns operation transforms each column of four bytes using a special mathematical function. This function takes as input the four bytes of one column and outputs four completely new bytes using left shift and XOR. The AddRoundKey operation XORs the state matrix to the round key.

Overall, the basic operations in three encryption algorithms can be classified into conventional arithmetic addition (+), modulo 2 addition (XOR, AddRoundKey, MixColumns), logical shift (rotate left, ShiftRows, MixColumns), permutation operation (EP-box, SP-box), and substitution box (S-box). To realize the parity check-based fault detection approach, a parity prediction method should be designed for each of the basic operations, which will be presented in Section 5.

## 3. Threat Model

The threat model considered in this paper is that attackers inject timing faults into cipher circuits with the aim of revealing the encryption key, by means of FIAs such as clock glitches, electromagnetic pulse, and voltage underfeeding.

In a sequential logic path, data launched from $Reg_1$ is propagated through the combinational logic and is captured by $Reg_2$. For correct operation, the following setup timing requirement should be met.

$$T_{clk} \geq T_{pd} + T_{su} \qquad (2)$$

where $T_{clk}$ is the clock cycle, $T_{pd}$ is the signal propagation delay through the combinational logic, and $T_{su}$ is the setup time. The FIAs such as the clock glitch attack and electromagnetic pulse attack could produce clock glitches in the clock signal, which is equivalent to temporarily decreasing $T_{clk}$. The FIA technique voltage underfeeding could increase the signal propagation delay $T_{pd}$. In these two cases, the setup timing constraint (2) could be violated, and $Reg_2$ samples the wrong data, resulting in faulty operation [18]. It is reported in [8] that for clock glitch attack, as the fault intensity increases (i.e., the glitch period decreases), more bits fail one after another.

With the means of timing fault injection, the attackers usually choose the appropriate time and position to induce faults, in order to reveal key with the least effort. For example, when applying FIA to the AES circuit, faults are injected into the state bytes in the SubBytes operation at the beginning of the last round [19,20], or into the operations before MixColumns in the second/third last round with more complex key analysis [21]. Therefore, we define the encryption operations in a specific round, which are particularly targeted by FIA, as the *security-critical operations*.

There are usually two ways of implementing a cipher circuit. For small hardware areas, the cipher is implemented in an iterative way, i.e., different rounds of operations are mapped on the same piece of a circuit. For high speed, the cipher is implemented in pipeline, and there is a piece of the circuit for each round of operations. Figure 2 shows an example, where AES is implemented in two ways. By this example, we want to show how the security-critical operations (shaded blocks) are defined for different implementations. Note that, the example just shows one possible selection of security-critical operations. Circuit designers could have different specifications, e.g., selecting as the security-critical operations all operations in the last round in Figure 2b.

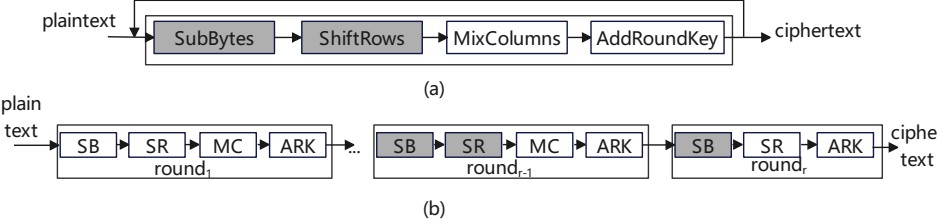

**Figure 2.** The example of AES circuit in (**a**) iterative implementation and (**b**) pipelined implementation. Shaded blocks represent security-critical operations.

In this work, to balance fault coverage and hardware overhead, the fine-grained parity check is applied to the security-critical operations, while the coarse-grained parity check is applied to the other operations. The proposed approaches are presented in the next section.

## 4. Proposed Approaches

As discussed early, high fault coverage needs fine-grained parity check but introduces high hardware overhead. Considering the nature of FIAs, faults are more likely to be injected into the security-critical operations defined in this paper. In other words, faults injected into the other operations are not the main target of the proposed detection approaches. This is one of the main differences between security-driven design and fault tolerance-driven design. Therefore, this paper proposes two parity check-based detection approaches against timing FIAs: mixed-grained parity check approach (Section 4.1) and word recombination parity check approach (Section 4.2).

### 4.1. Mixed-Grained Parity Check

**Concept and Implementation:** Based on the fact that fine-grained parity check achieves higher fault coverage but introduces higher hardware overhead, it is intuitive to apply fine-grained parity check to the security-critical operations and coarse-grained parity check to other operations in cipher circuits. The proposed mixed-grained parity check approach is based on this idea.

Figure 3a shows an example of the mixed-grained parity check, where $Operation_0$ and $Operation1$ process 32-bit data and $Operation_0$ is assumed to be the security-critical operation. 32-bit parity check is applied to $Operation_1$, and a parity check block (PCB) is added. Given the parity check nature, the theoretical fault coverage is 50%. Then, a 16-bit parity check is applied to $Operation_0$. The 32-bit $A_0$ is divided into two 16-bit banks, and a PCB is added for each bank. In this way, 2 parity bits are calculated, which increase the theoretical fault coverage to 75%, at the cost of 2 PCBs. Therefore, the mixed 16/32-bit parity check consumes 3 PCBs and has the $50 \sim 75\%$ fault coverage, as shown in Table 1. In practice, the $Operation$s of the pipelined or iterative cipher circuits can be the cipher rounds (pipelined cipher circuits) or the basic operations in one cipher round (iterative cipher circuits). Figure 4 shows the operation division of DES as a practical example. In the figure, each Operation of the pipelined DES means one DES cipher round, and one $Operation$ of an iterative DES means one operation in one cipher round.

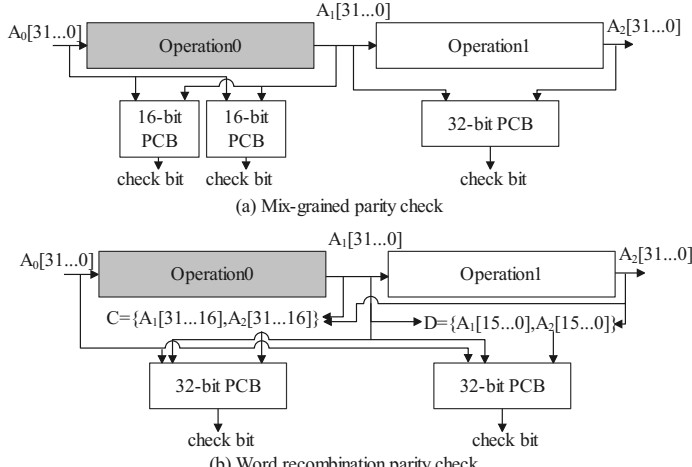

**Figure 3.** The implementation of mixed-grained parity check and word recombination parity check. $Operation0$ is the security-critical operation. $C$ and $D$ are new words recombined by $A_1$ and $A_2$. PCB: parity check block.

**Table 1.** Theoretical analysis of fault coverage and hardware resource usage of different parity check approaches.

| Granularity $m$ | Uniform 32-bit | Uniform 16-bit | Mixed 16/32-bit | 2 Words Recombination |
|---|---|---|---|---|
| Fault coverage (%) | 50 | 75 | $50 \sim 75$ | 75 |
| HW resource | 2 PCB | 4 PCBs | 3 PCBs | 2 PCBs |

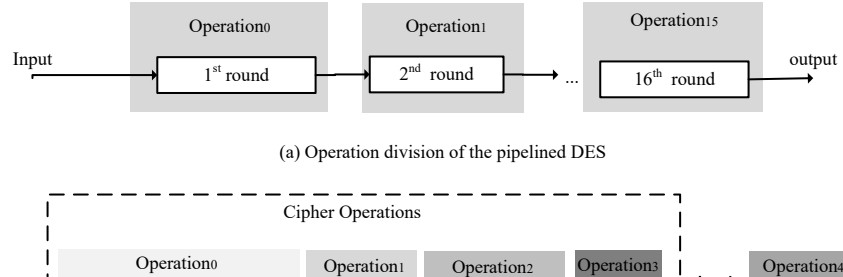

(a) Operation division of the pipelined DES

(b) Operation division of the iterative DES

**Figure 4.** The example of *Operation* division.

### 4.2. Word Recombination Parity Check

To further reduce the hardware overhead, we also propose a word recombination parity check approach. The approach essentially exploits the parity check resources assigned for the security non-critical operations to achieve fine-grained parity check for the security-critical operations. Specifically, words from the security-critical operation and the non-critical operation are partitioned into sub-words and the sub-words are recombined to form new words whose parity is checked at the word level. Next, the approach is presented in detail.

**Concept:** Figure 5 illustrates the concept of word recombination parity check. In Figure 5a 32-bit word $A$ and word $B$ are partitioned into two 16-bit sub-words $\{A1, A2\}$ and $\{B1, B2\}$, respectively. Assume that parity of $A$ and $B$ is checked with 32-bit, respectively. Then, new words $C = \{A1, B1\}$ and $D = \{A2, B2\}$ are formed and their parity is also checked with 32-bit. Assuming that two 1-bit faults occur in $A1$ and $A2$, respectively, and no fault exists in $B$. The 32-bit parity check of $A$ cannot detect the faults because the number of faults is even. After recombination, the 32-bit parity check of $C$ and $D$ can detect the faults without introducing extra parity bits. This is equivalent to 16-bit parity check $A$. A fine grained parity check can be achieved by recombining $A$ with more words. For example, partitioning $A$ into 4 8-bit sub-words and recombining with other three words can achieve 8-bit parity check for $A$.

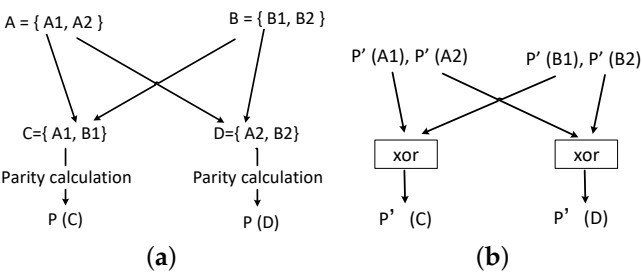

**Figure 5.** The example of the word recombination parity check. (**a**) Recombination of words; (**b**) Recombination of parities.

**Implementation:** To implement the idea of the word recombination parity check, we need to (1) find the words for recombination and (2) realize the parity prediction and the parity calculation of the recombined words.

The parity calculation of the recombined words still uses (1), as shown in Figure 5a. The parity prediction of the recombined words can be implemented in two steps. The first step predicts the parity of each sub-word according to the related operation. The second step generates the parity of the recombined words by XORing the predicted parities of the sub-words. Continue with the example in Figure 5. The predicted parity means $P'(C) = P'(A1) \oplus P'(B1)$ and $P'(D) = P'(A2) \oplus P'(B2)$, respectively, as shown in Figure 5b.

The last requirement to apply the word recombination approach is to find proper words for recombination. Given a word from a security-critical operation, there are two cases. For the iterative circuit implementation of ciphers, the words can be from the security non-critical operations within the same round. For the pipelined circuit implementation, the words can be from the security non-critical operations within the same round and other rounds. Note that the data dependency between words does not affect the word recombination because the parity prediction is carried out in each sub-word individually.

Figure 3b shows an example of the word recombination parity check. $A_1$ and $A_2$ are recombined to form $C$ and $D$. As shown in the figure, $A_0$ and $A_1$ are used to predict the parity of $C$ and $D$, respectively. Parity calculation is carried out on $C$ and $D$ directly. As a result, by exploiting the 32-bit PCB of *Operation*1, a 16-bit parity check is achieved for *Operation*0. Therefore, the 2-word-recombination parity check consumes 2 PCBs and has the 75% fault coverage, as shown in Table 1. Note that, in this example, there is a data dependency between $A_1$ and $A_2$. However, as long as both $A_0$ and $A_1$ have valid values at the same time, the approach works. In circuit design, this requires $A_0$, $A_1$, and $A_2$ to be registered.

Although reducing resource usage, the word recombination approach could affect the timing of the circuit because it introduces a logical relationship between the words and may affect their placement and routing. The experiment in Section will evaluate this effect.

So far, we have presented two parity check-based FIA detection approaches. As shown in Figure 1, the PCB contains parity calculation, prediction, and comparison. As described, while the parity calculation and comparison are easy to implement, implementation of the parity prediction depends on the specific operations in encryption algorithms and is not straightforward. The next section presents the parity prediction methods of different basic operations introduced in Section 2.2.

## 5. Parity Prediction of Basic Operations

Ideally, parity prediction predicts the parity of an operation's output based on its input's parity, without duplicating the real operation of CUA. In this way, the high overhead of dual modular redundancy is avoided. However, different basic operations show different features, such as some are linear, and some are nonlinear. This diversity increases the design difficulty of parity prediction. This section will present the parity prediction methods for the basic operations introduced in Section 2.2. Because these basic operations are common to existing encryption algorithms, the presented prediction methods enable wide applications of the proposed parity check-based approaches.

### 5.1. Conventional Arithmetic Addition

For operation $A$ adding $B$, its parity $P(A + B)$ can be predicted by (3)

$$P(A + B) = P(A) \oplus P(B) \oplus C_{in} \oplus C_{out}^{(i)} \tag{3}$$

where $C_{in}$ is the carry input and $C_{out}^{(i)}$ is the carry generated by $A[i] + B[i]$ ($0 \leq i \leq n - 2$). We can see that as long as inputs $A$, $B$ and $C_{in}$ are known, $P(A + B)$ can be obtained by operating on the parities of $A$ and $B$, instead of $n$-bit $A$ and $B$.

### 5.2. Modulo 2 Addition

The parity prediction for modulo 2 operation is straightforward. The parity of *A* xor *B* can be obtained as below

$$P(A \oplus B) = P(A) \oplus P(B) \tag{4}$$

### 5.3. Logical Shift

As described in Section 2, the logical shift operation includes left rotation ($<<<$) and left shift ($<<$). Given a *n*-bit word *A* and after *k*-bit left rotation, the operation does not change the parity of *A*, i.e., $P(A <<< k) = P(A)$. However, when fine-grained parity check is applied, the rotate shift does change the parities of different sub-words of *A*. For *m*-bit parity check, the parity of the *h*-th sub-word of *A* after *k*-bit left rotation is predicted by (5).

$$P_h(A <<< k) = \overset{i=(m-1)\sim 0}{\oplus} A[(mh - k + i) \textbf{ mod } n] \tag{5}$$

For left shift, the parity of the *h*-th sub-word of *A*, with *m*-bit parity check, after *k*-bit left shift is predicted by

$$P_h(A << k) = \overset{i=(m-1)\sim 0}{\oplus} A[(mh - k + i)], \; mh - k + i \geq 0 \tag{6}$$

### 5.4. Permutation Operation

The output of the permutation operations is a permutation of the input according to a permutation table. Figure 6 illustrates the permutation operation of EP-box and SP-box. We can see that given the fixed permutation logic and input *A*, the parity of output *B* with different granularity can be predicted by XORing the corresponding bits of *A*. For example, applying 4-bit parity check to the SP-box operation in Figure 6b, $P_7(\text{SP-box}(A)) = A[24] \oplus A[3] \oplus A[10] \oplus A[21]$ and the parities of other seven sub-word can be predicted similarly.

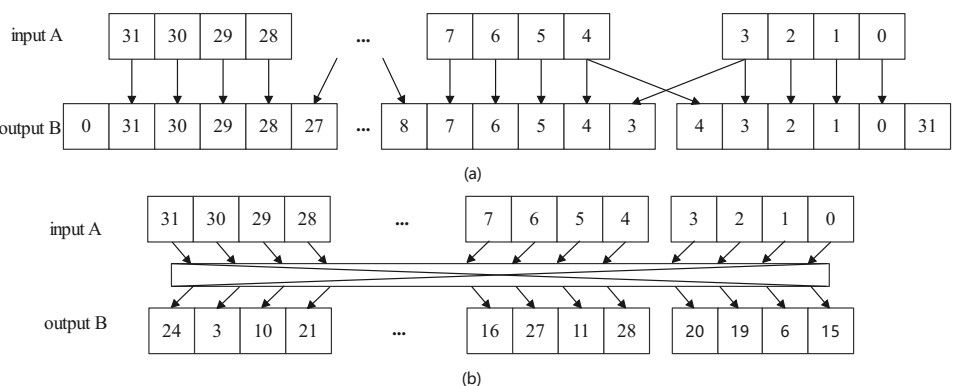

**Figure 6.** Permutation operation of (**a**) EP-box and (**b**) SP-box.

### 5.5. Substitution Box

The S-box operation carries out the real mixing and the outputs are generated by looking up a fixed table (state matrix) in AES or eight fixed tables in DES. As a result, the parity of the S-box operation output cannot be predicted using the parity of the input. To solve the problem, a separate parity table, which includes the predicted parity bit for each element of S-box, is built. Then, the predicted parity bit can be obtained by looking up the parity table.

Figure 7 illustrates the parity prediction of S-box and ShiftRows in AES. The state matrix in Figure 7a is the S-box table. Its parity table in a form of a matrix is shown in Figure 7b, where $p_{r,c}$ is the parity of $s_{r,c}$. Given an input, both the state matrix and parity matrix are looked up, and the outputs are used in parity calculation and prediction,

respectively. Note that, in Figure 7 each element of the state matrix is 8-bit. If a coarse-grained parity check is used, the corresponding elements of the parity matrix are XORed to form the required parity. If the fine-grained parity check is applied, the parity matrix can be expanded, and each element indicates the parity of different parts of the elements of the state matrix. For example, applying the 4-bit parity check adds two parity bits for each state matrix element, i.e., $16 \times 2$ parity bits in total. In addition, with the parity matrix, the parity of ShiftRows operation result is easy to obtain, instead of using (5).

$$
\begin{array}{cccc}
s_{0,0} & s_{0,1} & s_{0,2} & s_{0,3} \\
s_{1,0} & s_{1,1} & s_{1,2} & s_{1,3} \\
s_{2,0} & s_{2,1} & s_{2,2} & s_{2,3} \\
s_{3,0} & s_{3,1} & s_{3,2} & s_{3,3} \\
\end{array}
\qquad
\begin{array}{cccc}
p_{0,0} & p_{0,1} & p_{0,2} & p_{0,3} \\
p_{1,0} & p_{1,1} & p_{1,2} & p_{1,3} \\
p_{2,0} & p_{2,1} & p_{2,2} & p_{2,3} \\
p_{3,0} & p_{3,1} & p_{3,2} & p_{3,3} \\
\end{array}
\qquad
\begin{array}{cccc}
p_{0,0} & p_{0,1} & p_{0,2} & p_{0,3} \\
p_{1,1} & p_{1,2} & p_{1,3} & p_{1,0} \\
p_{2,2} & p_{2,3} & p_{2,0} & p_{2,1} \\
p_{3,3} & p_{3,0} & p_{3,1} & p_{3,2} \\
\end{array}
$$

(a)          (b)          (c)

**Figure 7.** Parity prediction of S-box and ShiftRows in AES. (**a**) State matrix; (**b**) Parity matrix; (**c**) Parity matrix after ShiftRows.

With the parity prediction methods of the above basic operations, the proposed parity check approaches can be applied to various encryption circuits. The next section will present the application flow of the proposed approaches.

## 6. Design Flow

This section explains the design flow of the proposed approaches, which can be integrated into the existing IC design flow. Figure 8 shows the design flow. The input to the design flow is a data flow graph (DFG) of CUA. Each node of DFG represents an encryption operation of an encryption algorithm, and an edge indicates the input-output relationship between operations.

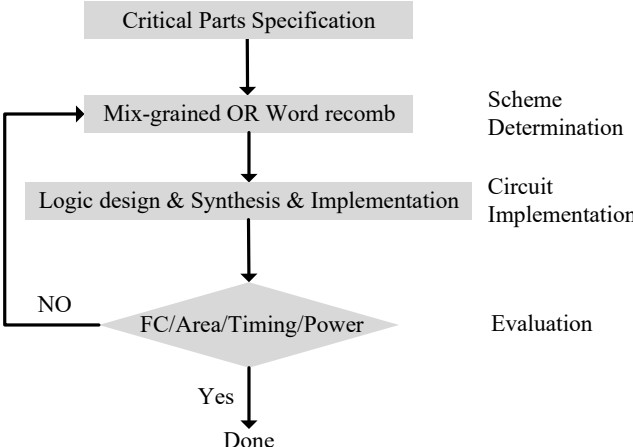

**Figure 8.** The design flow of the proposed approaches. In the Evaluation stage, *FC* means fault coverage.

Designers should specify area/timing/power/security requirements and security-critical operations. With the provided information, the design flow explores the design space in three stages: Scheme determination, Circuit implementation, and Evaluation.

### 6.1. Critical Parts Specification

In the specification stage, a designer should specify the security-critical parts of the main circuit according to the circuit's application. The last several cipher rounds of cipher circuits are specified as the security-critical parts by this work because faults in the rounds are easy to be used by attackers [22–24].

*6.2. Parity Check Scheme Determination*

The parity check scheme determination stage mainly decides the granularity of parity check, i.e., determining $m_i$ for each operation $i$. We formulate the decision process in a constrained optimization problem as below.

$$\max FC(m_0, m_1, \ldots, m_O) \tag{7}$$
$$s.t. Res(m_0, m_1, \ldots, m_O) \leq R^*$$

Here, the objective is to maximize the fault coverage $FC()$ with respect to resource constraint $Res()$. $R^*$ is the available hardware resource in a hardware platform. The next will show how the fault coverage and resource usage models are analytically formulated, respectively.

**The fault Coverage Model:** The fault coverage of mixed-grained parity check is dependent on parity check of each operation involved in a cipher circuit, especially the security-critical operations. Therefore, in this paper, we define $(FC(\cdot))$ as an average of the fault coverage $fc_i(m_i)$ of each operation $i$.

$$FC(m_0, m_1, \ldots, m_O) = \mathbf{Avg}\{\alpha_i fc_i(m_i)\} \tag{8}$$

where $\alpha_i$ is a weight associated with operation $i$. The security-critical operations will be assigned a large weight.

The fault coverage $fc_i(m_i)$ can be theoretically analyzed according to the used parity check granularity. Here, the fault coverage is defined as (9).

$$FC = \frac{Number\ of\ detected\ faults}{Total\ number\ of\ injected\ faults} \tag{9}$$

It is assumed that the number of faulty bits appears randomly in a word. Thus, the total number of the $i$-bit faults for an $n$-bit word is $C_n^i$, and the total number of faults is $\sum_{i=1}^{n} C_n^i$. Table 2 shows the theoretical fault coverage of parity check with $m = 32, 16, 8, 4$, respectively, for a 32-bit word. For example, the 32-bit parity check uses one parity bit and in theory can detect the faults with the odd number of bits but not the even number of bits, and thus the fault coverage is 50%. The 16-bit parity check uses two parity bits and in addition to all odd-bit faults can also detect the even-bit faults in which the number of faulty bits in each 16-bit sub-word is odd. As a result, the theoretical fault coverage is about 75%. Similarly, the theoretical fault coverage for 8-bit and 4-bit parity check can be derived. By looking up the table, $fc_i(m_i)$ can be obtained given $m_i$.

**Table 2.** Theoretical fault coverage of different degrees of parity check granularity for 32-bit word.

| $m_i$ | 32-bit | 16-bit | 8-bit | 4-bit |
|---|---|---|---|---|
| $fc_i$ (%) | 50 | 75 | 93.75 | 99.55 |

**The resource Model:** As shown in Figure 1, each parity bit needs a PCB. The resource overhead is comprised of three blocks: parity prediction, parity calculation and comparison. The required logic circuit to implement the three blocks mainly includes XOR gates and registers. Therefore, we use the number of XOR gates (#XOR) and the number of registers (#Reg) to evaluate the resource usage.

Among the three blocks, the prediction procedure for different operations is different, and thus the resource usage of the prediction block varies over operations and parity check granularity $m$. Based on the prediction methods presented in Section 5 for the basic operations, the resource usage is listed in Table 3. The resource usage of the parity calculation and comparison blocks only depends on $m$. The detailed derivation procedure is omitted for brevity.

**Table 3.** Resource usage of parity check approach.

| Blocks | | #XOR | #Reg |
|---|---|---|---|
| Prediction | Addition | $(m-1)n/m$ | $3n/m$ |
| | Modulo 2 addition | $n/m$ | $3n/m$ |
| | Logical shift | 0, if $m = n$; $(m-1)n/m$, if $m < n$; | $n/m$ |
| | Permutation | $(m-1)n/m$ | $n/m$ |
| | AES S-box | 3, if $m = 32$; 1, if $m = 16$; 0, others | $32 + n/m$, if $m = 4$; $16 + n/m$, others |
| Calculation+Comparison | | 32 | $2n/m$ |

Having the resource usage model for each block and operation, the total resource usage introduced by the proposed parity check approaches can be determined according to the encryption operations of CUA. The resource constraint must be met for all two resource types, i.e., $Res_{XOR}$ and $Res_{Reg}$ are formulated respectively.

The formulation in (7) is a $O$-variable integer non-linear programming problem. A simulated annealing algorithm can be used to obtain a near-optimal solution $(m_0, m_1, \ldots, m_O)$. Note that $m_i \leq n$ and usually takes value of power of 2. Also, $m_i = 0$ is allowed, meaning no parity check for operation $i$.

After determining the parity check granularity for each operation, the proposed two approaches can be applied. Considering application conditions and efficiency, it is suggested that word recombination can be applied first. If the word recombination approach cannot be used in certain conditions such as not finding the proper words for combination or the recombination introduces timing violations in the performance evaluation stage, then the mixed-grained approach is applied. Note that two approaches can be used together.

*6.3. Circuit Implementation*

After the parity check scheme is determined for CUA, the proposed approaches are applied to CUA and add PCBs at the RTL level with the basic design methodology. Based on the design methodology, a circuit is advised to be designed in hierarchies for the convenience of subsequent debugging, analysis, and optimization. Then, the default Synthesis & Implementation transforms the RTL-specified design into a gate-level representation, and places & routes the netlist onto device resources within the logical, physical, and timing constraints of the design. Finally, the corresponding bitstream is generated by design tools. This basic design flow is suitable for the implementation on FPGA and ASIC.

*6.4. Evaluation*

Evaluation stage evaluates *FC* (fault coverage) and performance of the circuit design.

*FC* **Evaluation:** As description in Section 3, attackers attempt to inject timing faults into CUA by inducing timing violations. To evaluate *FC*, we can mimic timing fault injection during post-layout timing simulation. This can be easily achieved by adding clock glitches into the clock signal during timing simulation, as shown in Figure 9. The clock signal *clk* with a glitch is generated by two clock signals with different clock cycles under the control of the selection signal *sel*. *clk_slow* represents the normal clock signal. The clock cycle of *clk_fast* is proportional to *clk_slow*, e.g., the proportion is 1/4 in Figure 9. By controlling the time when *clk_fast* signal is selected, timing faults can be injected into the encryption operations.

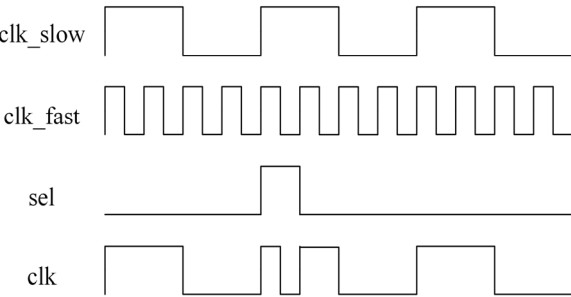

**Figure 9.** Clock glitch for mimicking timing fault injection.

After the timing fault injection simulation, the *FC* of the circuit design can be evaluated by counting the number of detected faults. If the fault coverage does not meet the security requirement specified by designers, then the flow goes back to the first stage to modify the parity check scheme. Note that, the *FC* in the first stage is theoretically estimated, and thus it is possible that the security requirement is not met. If the security requirement is met, the flow goes to the Performance evaluation.

**Performance Evaluation** evaluates area, timing and power consumption with EDA tools and ensures the budget of CUA is not broken. If performance requirements specified by designers are not satisfied, the flow goes back to the first stage. When back to the first stage, the optimization problem will be searched for an optimal solution or will be updated with different weights $\alpha_i$ and constraints, depending on the feedback from the second and third stages. Then the parity check scheme will be changed using different granularity, approaches or combinations of the proposed parity check approaches. Designers having deep knowledge about the encryption circuits can control the convergence of the design flow by setting proper security and performance requirements. The above design flow can be integrated with the existing IC design tool chain, and enables the design automation of the proposed parity check approaches. The next section will evaluate the proposed approaches.

## 7. Experiment Results

### 7.1. Experiment Setup

To evaluate the performance and efficiency of the proposed approaches, experiments are carried out on a Xilinx v7-690 FPGA board using the Vivado development environment. Three encryption algorithms AES, DES and RC5 are implemented on FPGA. AES and RC5 are implemented in the pipelined circuit, and DES is implemented in both iterative and pipelined circuits.

As described in Section 6, post-layout dynamic timing simulation is carried out to evaluate the detection efficiency. In experiments, a fault is injected if the circuit output is wrong, and the fault number is obtained by comparing the circuit output, and the correct ciphertexts [19]. In experiments, the working frequency of the cipher circuits (*clk_slow*) is 50 MHz. *clk_fast* means the injected clock glitches. The plaintexts are random, and the clock glitch attack begins after 100 *clk_slow* periods. In a clock glitch attack, the first faulty bit is on the critical path, which has the maximum path delay; then, as the glitch period decreases (i.e., the fault intensity increases), more bits fail one after another depending on the length of each bit's propagation path delay. Therefore, experiments in this work inject glitches with various periods by decreasing the *clk_fast* period to evaluate the detection efficiency comprehensively. In experiments, *clk_fast* starts with a period of 5*ns* and ends with 1*ns* at the rate of 1*ns* per step. At each step, 200 glitches are injected, and each glitch is injected every 40 *clk_slow* periods. As a result, each experiment injects 1000 glitches.

In the circuit design, multiple PCBs operate independently and all check bits are ORed to generate an alarm output. Alarm output equal to 1 indicates that injected fault is detected.

### 7.2. Results

7.2.1. Evaluation of Mixed-Grained Parity Check

**The pipelined cipher circuits:** Figures 10–12 show the overhead and the detection rate (i.e., *FC*) of different mixed-grained parity check schemes for RC5, pipelined DES, and AES, respectively. The *x*-axis represents various mixed-grained schemes.

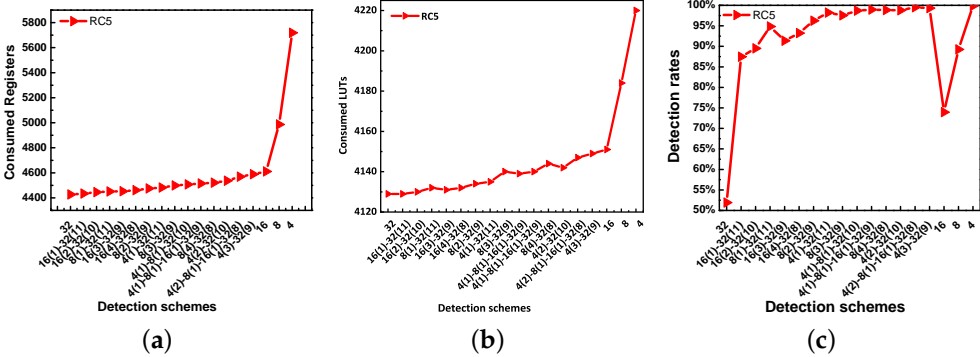

**Figure 10.** The results of RC5 with various mixed-grained schemes. (**a**) Register consumption; (**b**) LUT consumption; (**c**) Detection rate. The detection rate here means *FC*.

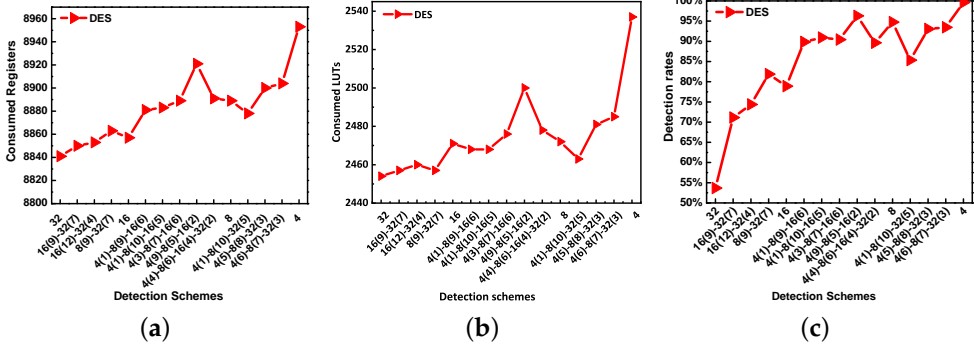

**Figure 11.** The results of the pipelined DES with various mixed-grained schemes. (**a**) Register consumption; (**b**) LUT consumption; (**c**) Detection rate (*FC*).

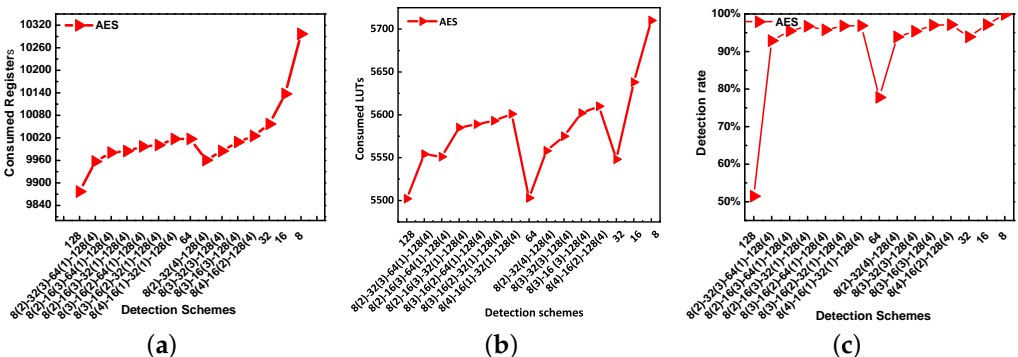

**Figure 12.** The results of AES with various mixed-grained schemes. (**a**) Register consumption; (**b**) LUT consumption; (**c**) Detection rate (*FC*).

In the experiments, the ciphers contain 10 (AES), 16 (pipelined DES), and 12 (RC5) cipher rounds. Therefore, the corresponding circuit is composed of 10 (AES), 16 (pipelined DES), or 12 (RC5) cipher blocks. In the figures, the mixed-grained schemes are named by the grain of the cipher blocks. To describe the mixed-grained schemes, the blocks are in descending order based on path delay. For example, the mixed scheme 4(1)-8(1)-32(10)

means that 4-bit PCB is applied to the first block, 8-bit PCB is applied to the second block, and 32-bit PCBs are applied to the rest ten blocks. For another example, the mixed scheme 4 means the 4-bit PCB is applied to each block of the CUA.

To obtain the optimal mixed-grained scheme, various mixed-grained schemes are evaluated by experiments. Results in the figures show that register consumption, LUT (Look Up Table) consumption, and the *FC* increase with the number of parity bits. It is shown that the increased register consumption is greater than LUT consumption, which is consistent with the analysis of Table 3. Therefore, the area evaluation mainly discusses the increased register consumption. To identify the optimal scheme for the mixed-grained approach, the efficiency of the mixed-grained scheme (*ES*) in (10) is introduced. The higher *ES* means the higher *FC* and fewer registers consumption for the mixed-grained scheme. In other words, the mixed-grained scheme is efficient against the attack. The optimal scheme means the mixed-grained scheme with the greatest *ES*. In (10), $\alpha$ or $\beta$ is the weight of *FC* or register consumption which can be adjusted by the designer based on the requirement. In this paper, $\alpha$ and $\beta$ are set to 1, and the results of *ES* are shown in Figure 13.

$$ES = \frac{\alpha \cdot FC \cdot 100}{\beta \cdot Register\ consumption} \tag{10}$$

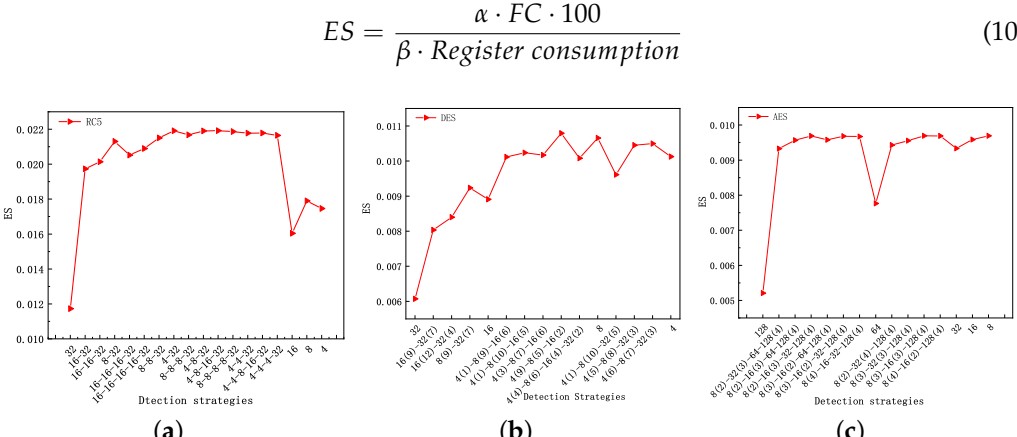

**Figure 13.** *ES* of the mixed schemes for (**a**) RC5, (**b**) pipelined DES, and (**c**) AES.

In Figure 13a, the greatest *ES* of RC5 is 0.217 belonging to 4(2)-8(1)-16(1)-32(8). Therefore, the optimal scheme in RC5 experiment is 4(2)-8(1)-16(1)-32(8). Compared with the 32-bit scheme in Figure 10, the optimal scheme increases the FC by 51.92% and only consumes 3.2% more registers. Compared with the 4-bit scheme, the optimal scheme saves 20.1% registers and has a 0.35% lower *FC*. Figures 11 and 13b show the results of pipelined DES. It is shown that the greatest *ES* is about 0.011 which belongs to 4(9)-8(5)-16(2). The optimal scheme 4(9)-8(5)-16(2) has a 96.29% *FC* which is 53.69% higher than the 32-bit scheme, at the cost of 0.9% more registers. Similarly, the result of AES is shown in Figures 12 and 13c. For AES, 8(3)-16(3)-128(4) is the optimal scheme with 0.098 *ES*. All these results demonstrate the high efficiency of the proposed mixed-grained parity check scheme for the three ciphers.

Table 4 shows the overhead of cipher circuits with no detection or optimal mixed schemes. WNS (Worst Negative Slack) means the worst difference between the actual and the target delay. The negative WNS means the setup timing violation. The Δ data is calculated by comparing the optimal mixed-grained schemes with the circuit without detection. Results show that the mixed-grained method has little influence on timing performance compared with the circuits without detection, which benefits from the integration at the RTL level. Also, the small circuit size of the PCBs results in limited overhead on power consumption and hardware resources.

**Table 4.** Overhead of the circuits with no detection or optimal mixed schemes.

| | Implementation | Reg | Power (W) | WNS (ns) | ΔPower (W) | ΔWNS (ns) | ΔReg |
|---|---|---|---|---|---|---|---|
| AES | No detection | 9657 | 0.686 | 6.368 | - | - | - |
| | 8(3)-16 (3)-128(4) | 10009 | 0.708 | 6.228 | 0.022 | 0.14 | 352 |
| Pipelined DES | No detection | 8574 | 0.452 | 7.065 | - | - | - |
| | 4(9)-8(5)-16(2) | 8920 | 0.63 | 6.932 | 0.178 | 0.133 | 346 |
| RC5 | No detection | 4351 | 0.57 | 3.317 | - | - | - |
| | 4(2)-8(1)-16(1)-32(8) | 4524 | 0.668 | 3.192 | 0.098 | 0.125 | 174 |

**The iterative cipher circuit:** This experiment evaluates the efficiency of the mixed-grained method for the iterative DES implementation. The iterative DES with mixed schemes is implemented as Figure 4. Table 5 shows the results. 4(1)-32(3) is the optimal scheme with $ES = 1.351$. $FC$ of 4(1)-32(3) is 98.6% with 73 registers. Compared with 4(4), 4(1)-32(3) is 0.7% lower in $FC$ while saving 21 registers. Compared with 32(4), 4(1)-32(3) consumes 7 registers more while being 44.7% higher in $FC$. Compared with the iterative DES with no detection, iterative DES with the optimal mixed scheme only increases 17 registers, 48 LUTs, 0.03 W power, and 0.2027 ns WNS.

**Table 5.** Results of mixed schemes with the iterative DES.

| Mixed Scheme | FC | Reg | LUT | ES | Power (W) | WNS (ns) |
|---|---|---|---|---|---|---|
| No detection | 0 | 56 | 92 | 0 | 0.3 | 4.958 |
| 32(4) | 53.9% | 66 | 135 | 0.817 | 0.33 | 4.798 |
| 4(4) | 99.3% | 94 | 153 | 1.056 | 0.331 | 4.68 |
| 4(1)-32(3) | 98.6% | 73 | 140 | 1.351 | 0.33 | 4.7553 |

7.2.2. Evaluation of Word Recombination Parity Check

**The pipelined cipher circuits:** To evaluate the efficiency of the word recombination approach, it is applied to AES, pipelined DES, and RC5 encryption circuits. In the experiment, the parity check with one parity bit is applied to all cipher blocks.

The results are shown in Table 6. LUT consumption is not shown because it is the same as the parity check with one parity bit. *Num of words* means the number of recombined words. When *Num of words* $= 1$, there is no recombination operation. In Table 6, it is shown that the $FC$s with word recombination are significantly improved, while the register consumption is limited. For example, when 8-word-recombination $FC$ is increased by 46.29% (RC5), 47.16% (pipelined DES), and 44.5% (AES) compared to the 1-word-recombination scheme, while register consumption is only increased by 0.79% (RC5), 2.35% (pipelined DES), or 0.588% (AES), respectively. At the same time, the word recombination has little overhead on power consumption and WNS as shown in Table 6.

Furthermore, the word recombination approach with 8 recombined words is compared with the optimal scheme of the mix-grained parity check. For RC5, the optimal scheme of the mix-grained parity check is 4(2)-8(1)-16(1) whose $ES = 0.217$, while 8-word-recombination $ES$ is 0.022. For DES, 8-word-recombination $ES$ is 0.011, and the optimal mix-grained scheme is 4(9)-8(5)-16(2) with 0.01 $ES$. For AES, 8-word-recombination $ES$ is 0.0098, and the optimal mix-grained scheme is 8(3)-16(3)-128(4) with 0.0097 $ES$. These results mean that 8-word- recombination scheme is more efficient than the optimal scheme of the mix-grained approach.

**Table 6.** Results of word recombination.

| Circuit | Num. of Words | FC | Reg | ES | Power (W) | WNS (ns) |
|---|---|---|---|---|---|---|
| RC5 | 1 | 51.92% | 4427 | 0.012 | 0.61 | 3.252 |
| | 2 | 87.48% | 4432 | 0.019 | 0.63 | 3.25 |
| | 4 | 94.83% | 4443 | 0.021 | 0.68 | 3.174 |
| | 8 | 98.21% | 4462 | 0.022 | 0.705 | 3.117 |
| Pipelined DES | 1 | 53.69% | 8837 | 0.006 | 0.483 | 6.982 |
| | 2 | 78.54% | 8841 | 0.009 | 0.505 | 6.978 |
| | 4 | 89.57% | 8857 | 0.010 | 0.527 | 6.953 |
| | 8 | 98.19% | 8889 | 0.011 | 0.54 | 6.942 |
| AES | 1 | 51.44% | 9877 | 0.0052 | 0.697 | 6.368 |
| | 2 | 75.11% | 9965 | 0.0075 | 0.743 | 6.302 |
| | 4 | 90.83% | 9997 | 0.0091 | 0.78 | 6.221 |
| | 8 | 98.60% | 10109 | 0.0098 | 0.824 | 6.115 |

**The iterative cipher circuit:** This experiment evaluates the efficiency of the word recombination method with one iterative DES. Table 7 shows the result of the word recombination and *ES* increases with the number of recombined words. The 4-word recombination scheme is the optimal scheme with $ES = 1.059$.

**Table 7.** Results of word recombination with the iterative DES.

| Num. of Words | FC | Reg | ES | Power (W) | WNS (ns) |
|---|---|---|---|---|---|
| 1 | 53.9% | 66 | 0.817 | 0.33 | 4.798 |
| 2 | 74.2% | 74 | 1.003 | 0.332 | 4.723 |
| 4 | 84.7% | 80 | 1.059 | 0.339 | 4.69 |

Compared with 1-word recombination, the *FC* of 4-word recombination is 30.8% higher while consuming 14 registers more. Compared with the mixed scheme 4(1)-32(3), the 4-word-recombination *ES* is less than the ES of 4(1)-32(3). In Table 7, the 4-word-recombination scheme has limited timing overhead and power consumption compared with the 1-word-recombination scheme.

## 8. Discussion

This section summarizes the fault detection methods into two categories.

Sensor-based detection: The sensor-based detection is proposed to monitor the physical signals of the device, such as the clock or voltage. The typical methods include RO-based [25] or delay-chain-based detectors [26]. Generally, the sensor-based countermeasures have little influence on the protected circuits and bring low power and hardware overhead. In addition, the sensor-based countermeasures achieve high efficiency against specific faults. For example, the detector detects clock glitch attacks by monitoring the device's clock signal. As a result, the clock glitch monitor fails to detect the faults induced by increasing path delay. PV and aging phenomena also influence the efficiency of the detectors.

Redundancy countermeasures: The redundancy countermeasures detect data faults by redundancy (e.g., time, hardware, or information redundancy) and are independent of the attack technologies. Repetition is one typical time redundancy countermeasure with a high time overhead. Hardware redundancy detects faults by integrating duplicated or triplicated hardware with high hardware overhead [27]. Information redundancy detection detects faults by employing the error detection code (EDC) and consumes limited time and hardware overhead.

This work proposes parity-code-based methods with information redundancy. Compared with other EDC-based countermeasures [11,28,29], this work (1) abstracts the basic operations of typical ciphers and proposes the parity prediction for the basic cipher operations so that the proposed detection approach applies to multiple ciphers instead of one specific cipher; (2) realizes the idea of fine-grained parity check with low overhead in two ways, and provides designers with different design options; (3) develops a design flow, which can be integrated with the existing IC design flow, enabling security-driven hardware design flow.

## 9. Conclusions

This work proposes the mixed-grained parity check and word recombination parity check against timing FIAs and integrates the implementation flow of the two approaches with the existing IC design flow. The proposed methods can apply to multiple widely used cryptography ICs because this work designs parity check blocks for basic operations involved in various encryption algorithms. Evaluation of RC5, AES, and DES encryption implementations show that the mixed-grained approach increases the fault coverage rate by up to 53.69% while consuming 13.2% more resources compared with word parity check; the word recombination approach increases the fault coverage rate by up to 47.16% while introducing up to 2.35% resource usage. In the future, the proposed approaches will be explored to extend the application by designing parity check blocks for different circuits besides cryptography ICs.

**Author Contributions:** Conceptualization, P.W. and Q.L.; Methodology, M.Z., H.L. and P.W.; Software, M.Z.; Formal analysis, M.Z. and H.L.; Investigation, M.Z., H.L. and P.W.; Data curation, M.Z. and P.W.; Writing original draft, M.Z.; Writing review and editing, H.L. and Q.L.; Supervision, Q.L.; Project administration, Q.L.; Funding acquisition, Q.L. All authors have read and agreed to the published version of the manuscript.

**Funding:** This work is funded by the National Natural Science Foundation of China under Grant 61974102.

**Institutional Review Board Statement:** Not applicable.

**Data Availability Statement:** Data openly available in a public repository.

**Acknowledgments:** This work would like to thank the support of the National Natural Science Foundation of China under Grant 61974102.

**Conflicts of Interest:** The authors declare no conflict of interest.

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
