# Peer review of "Parity Check Based Fault Detection against Timing Fault Injection Attacks"

_electronics, doi:10.3390/electronics11244082_

Round 1
Reviewer 1 Report
The authors present two proposals against timing FIAs to be applied in cryptographic systems. These proposals are based on parity check and are able to detect errors provoked by changes in the clock frequency (whose origin can be external attacks -such as clock glitches, electromagnetic pulses or voltage underfeeding- or natural interferences).
The first proposal is based on applying (1-bit) parity check with different detail depending on the criticism of the protected word. In critical operations, the data word is split in subwords, and a (fine-grained) parity check is applied to each of these subwords. In non-critical operations, the whole word is protected with a (coarse-grained) parity check.
The second proposal is based on interleaving technique used in memories: data words in the same encryption round are protected with a coarse-grained parity check; however, these data words are generated by recombining subwords of the different words in the same round.
These proposals are applied to three classical encryption algorithms (i.e. RC5, DES and AES), obtaining different versions of each system depending on the size of the subwords in both proposals, and their error coverage and resorce usage are calculated.
The authors present these proposals as countermeasures against attacks, but I do not see any countermeasure if the only action introduced is a 1-bit parity check able to detect an odd number of changes in data words (or subwords), with no further reaction (such as generating a predefined invalid output, stopping the encryption process, or trying to correct the changes. As presented, the FIA would still success in "stealing" the encryption keys.
On the other hand, the authors do not calculate (or estimate) the fault coverage of word recombination parity check proposal. In my opinion, they could do applying to a couple of examples.
My third concern refers to "Experimental Results" section. In my opinion, it is quite poorly explained for several reasons:
* In the setup, it is not clear how many injections are performed per experiment (from the text, it seems that they simulate each model just once, injecting 1000 glitches, divided in five 200-glitches per step, having the system five steps), or how, when and why does the "clk_fast" increments its period. Also, the authors say that DES algorithm is implemented in two different circuits (an iterative and another pipelined), but oly one set of results is shown.
* Regarding the results, the authors refer to the efficiency of every circuit, but they do not show their values in any graph. In my opinion they should add a new graph showing this parameter for each encryption algorithm (and version) to make it easier for the reader the understanding of the results obtained.
* Also, I miss the values of the hardware expenses (registers and LUTs) of the algorithms implemented without any detection scheme. Table 4 compares power consumption and WNS of the circuits without and with detection schemes.
* Another inconsistence that I can find is the fact that the auhors do explain the number of cipher blocks of RC5 algorithm (i.e. 12), and how they split these blocks in terms of the length of the partial parity checks applied and to how many blocks. However, they do not include this description for DES and AES algorithms. This way, it is difficult to know if there is any possibility to compare the models implemented.
* The same concerns apply to the results of the word recombination models. In this case, moreover, the comparison is still more difficult to understand because they compare a system where all cypher blocks have been implemented in the same way (i.e. using one 32-bit word per operation, although recombined in different sub-word sizes) with the optimal mixed-grained parity check system in each algorithm.
Regarding the writing style of the paper, in my opinion it is poor, with a lot of syntax errors and misspellings. For instance:
1. The word "shit" appears several times in the text, instead of the right "shift".
2. The right expressions for the different detail levels are "fine-grained", "coarse-grained" and "mixed-grained". You use in the text other non-valid words, such as "finer-" (and "coarser-"), or "mix-".
3. There are incorrect verb tenses. For instance, on page 11 (line 351), you say "The resource constraint must met for all...". I guest you meant "The resource constraint must be met for all..."
4. In Fig. 8 (on page 10) you show the dessign flow of your approaches. In my opinion, the third stage should be named "Circuit implementation" instead of "Circuit implement". In fact, Section 6.3 is called this way, and not with the wrong name.
5. Sections 7.1 and 7.2 are very poorly written, and are hard to understand (e.g. what do "The 40 normal clock periods occur between the two injected glitches.", "...second to longest...", "The rest 10...", "...the weight-hand...", or "...(ES) in (10) is introduced..." mean?).
Please review extensively the whole text
Author Response
Please see the attachment of Reviewer#1.

Reviewer 2 Report
This work proposes two countermeasures against fault injection attacks: mix-grained parity check and work recombination parity checks. The proposed solution increases fault coverage with minimal overheads.
- The explanation and implementation details in Section 4.1 pertaining to the mix-grained parity check can be improved.
- State-of-the-art comparison and comparison with other non-parity check-based techniques (at least qualitatively) will make the paper interesting.
Author Response
Please see the attachment of the Reviewer#2

Reviewer 3 Report
The presentation of the paper is good and explained well. The author must improve the following,
1. Related work sections are missing, we recommend the author to include 5-6 recent year papers.
2. Figure 2: The author must produce the algorithm in standard format. Don't use figure format.
3. Create a separate section for discussion.
4. Novelty of the research should be highlighted.
Author Response
Please see the attachment of the Reviewer#3

Round 2
Reviewer 1 Report
Thank you for introducing all changes that I suggested. However, I still find some problems.
1.- First, and related to my first comment, I'm afraid that replacing literally "countermeasures" with "detection" is not a good idea. Note that "detection" has not a full meaning most of times; instead, it acts as an "adjective" (or complement), and needs another noun to modify or be modified by (for example: fault (or error) detection, or detection mechanism). So, by changing literally "countermeasures" with "detection" has made that some sentences or expressions are now meaningless (as an example, read carefully sentences in lines 5, 33, 35, 36, etc.). Probably, "fault detection" should be more correct in all these cases, as used in references [11-14]. On the other hand, "countermeasures" might be correct in other cases. So, please check each case individually, and use the right expression.
Anyway, my concern was particularly on the title of your paper, because no countermeasure is taken under fault detection (i.e. supposed timing attack). So, your paper (like [11-14]) actually proposes systems able to detect errors caused by timing attacks. And there is where "countermeasures" should be replaced.
2.- Regarding the fault coverage estimation of the 2-word-recombination parity check proposal, I do not see why it is between 50% and 75%. In my opinion, this scheme it is equivalent to uniform M/2-bit scheme (where M is the word length), so it should be 75%. Can you clarify why you say it is 50%~75%, please?
On the other hand, in the text you insist in the number of PCBs to compare the hardware resource usage. In my opinion this is an error, because (if I'm not wrong) the resources used by two 32-bit PCBs, or two 16-bit plus one 32-bit PCBs, or four 16-bit PCBs should be almost the same. That's to say, the hardware cost of the PCBs in all your approaches only would depend on the word length of the algorithm, and the overall hardware cost depends on other circuits. Can you please confirm that point?
3.- (Lines 219-224): In this paragraph you explain Fig. 5(a), but it is wrong. In the figure, A and B words are partitioned into {A1, A2} and {B1, B2}, respectively, and not {A0, A1} and {B0, B1}. And thus, D recombined word is composed by {A2, B2}, and not A0, B0. Surprisingly, C recombined word is correctly defined. Please correct text and/or the figure.
4.- Although you have answered my question related to the experiment setup, I still do not understand. I mean:
* It is not clear to me how many times you simulate each "model" to inject faults into it, being a model the implementation of each algorithm with each detection scheme. Let us assume model "32" of RC5 algorithm (i.e. the RC5 implementing 32-bit PCBs in all 12 blocks). How many times has been that model simulated (to be injected faults)? Is it a fixed quantity (with different random inputs), or it depends on the coverage obtained?
* Does "clk_fast" frequency change along the same model simulation, or in successive simulations?
* What does step mean in this context (see line 423)? Does refer to the encryption algorithms?
* If each model has been simulated multiple times, how has FC calculated? As the average of all simulations?
* If each model has been simulated only once, I do not see how the values of FC obtained are representative. If the results obtained from simulating a given model with different inputs could be different, why injecting just possibility?
Let me explain you that, although not being an expert in cryptography, I am very familiar with fault injection. So, if I do not understand the process, a non-familiar reader would have serious problems to understand the injection/attack process.
5.- (Page 15): Regarding the analysis of results of pipelined circuits:
* (Line 447): The greatest value of ES for unrolled DES is not 0.01; it is actually much closer to 0.011.
* (Line 450): No more comments for AES? Only the raw value of ES, with no connection to FC or the cost?
* (Line 455): Is it really necessary to explain here what does a negative value of WNS mean? There are no negative values in the tables, as all the models are assumed to fit in the selected FPGA. I would remove that sentence.
* (Table 4): Why the number of LUTs used is not included in the table, if you showed their values in graphs in Fig 10(b), Fig 11(b) and Fig 12(b)? And, consequently, add a new "'D'LUT" (where 'D' represents uppercase delta) column to compare the differences between the model implementing the original algorithm and the models with detection scheme. Also, I wonder why the "'D'WNS" values are not negative, if the values of WNS are lower in the models with detection scheme than in the models without detection scheme.
6.- You have included the results of iterative DES algorithm but, while you have included in your cover letter the data related to the original implementation with no detection mechanisms (in Table 4), you have not included in Table 5 of the manuscript. Could you please add to table, and also include the comparison of the three options with original circuit (by including "'D'LUT", "'D'Reg", "'D'Power", and "'D'WNS" columns like in Table 4).
7.- In my opinion, it would be interesting to discuss if it is more convenient to implement the model with the best ES (that you called "optimal") or one between the models having the best FC. Probably, by adding to Table 4 the results of the best ones (say top-3 or top-5), you could generate a nice discussion. Indeed, sometimes, it could be better to implement the optimal mechanism, while other times it might be better implementing one of the top-FC not being the best, to trade-off between performance and cost.
8.- In my opinion, the related works should have been included at the end of Section 2 Background. By the way, are not there any proposals using ECCs instead of EDCs?
9.- Finally, there are still some mistakes and/or typos:
* (Line 107): What is "CUT"? I guess it should be "CUA".
* (Line 123): "Algorithm 1 ~ Algorithm 3 shows ...". It's a plural, so it should be "show".
* (Lines 138-139): "AES In Algorithm 3, each round of AES contains ..." => Probably, it should be "In Algorithm 3, each round of AES contains ..."?
* (Lines 207, 243, 410, 411, 430, 471):"... pipeline cipher" and "... pipeline circuit..." => "... pipelined"?
* (Lines 225-226): "Finer gained ..." => "grained"
* (Line 411): "... and random plaintexts are input" => Maybe you meant "... and inputs are random plaintexts"?
* (Page 14, last line of unnumbered paragraph): "... are set as 1, ... Fig 13" => "... are set to 1, ... Fig 13."
* (Fig. 13): Graph size is different size from the graphs in figures 10 to 12. It is not important actually, but only for coherence with the other graphs. Graphs have different text font and size from the graphs in figures 10 to 12.
* (Table 4): Column "'D'Res" => "'D'Reg"?
